# Sampling site for SARS-CoV-2 RT-PCR—An intrapatient four-site comparison from Tampere, Finland

**Dominik Kerimov**[1]*, **Pekka Tamminen**[2,3], **Hanna Viskari**[2,4], **Lauri Lehtimäki**[4,5], **Janne Aittoniemi**[1]

1 Department of Clinical Microbiology, Fimlab Laboratories, Tampere, Finland, 2 Department of Internal Medicine, Tampere University Hospital, Tampere, Finland, 3 Department of Otorhinolaryngology–Head and Neck Surgery, Tampere University Hospital, Tampere, Finland, 4 Faculty of Medicine and Health Technology, Tampere University, Tampere, Finland, 5 Allergy Centre, Tampere University Hospital, Tampere, Finland

* dominik.kerimov@fimlab.fi, dominik.kerimov@tuni.fi

## Abstract

### Background

SARS-CoV-2 diagnosis relies on the performance of nasopharyngeal swabs. Alternative sample sites have been assessed but the heterogeneity of the studies have made comparing different sites difficult.

### Objectives

Our aim was to compare the performance of four different sampling sites for SARS-CoV-2 samples with nasopharynx being the benchmark.

### Study design

COVID-19 positive patients were recruited prospectively, and samples were collected and analysed for SARS-CoV-2 with RT-PCR from all four anatomical sites in 43 patients, who provided written informed consent.

### Results

All anterior nasal and saliva samples were positive, while two oropharyngeal samples were negative. There was no significant difference in the cycle threshold values of nasopharyngeal and anterior nasal samples while saliva and oropharynx had higher cycle threshold values.

### Conclusions

Anterior nasal swab performs as good as nasopharynx swab with saliva also finding all the positives but with higher cycle threshold values. Thus, we can conclude that anterior nasal swabs can be used for SARS-CoV-2 detection instead of nasopharyngeal swabs if the situation would require so.

Data are available from the Tampere University Hospital Ethical committee (contact via minna.maa.lahtinen@pshp.fi) for researchers who meet the criteria for access to confidential data.

**Funding:** This work was supported by funding from Tampere Tuberculosis Foundation; Research Foundation of the Pulmonary Diseases; Competitive State Research Financing of the Expert Responsibility area of Tampere University Hospital; Tampere University Hospital Support Foundation. Funders did not have any role in any part of the study. The funders had no role in study design, data collection and analysis, decision to publish, or preparation of the manuscript.

**Competing interests:** The authors have declared that no competing interests exist.

## Background

After the severe acute respiratory syndrome coronavirus 2 (SARS-CoV-2) was first identified in China in late 2019 [1], the procedure for its reverse transcription polymerase chain reaction (RT-PCR) detection was quickly shared and a robust diagnostic methodology was established [2]. Initially, nasopharyngeal swab sample for RT-PCR was the recommended sample. Subsequently alternative sample types have been assessed and saliva and anterior nasal swab have been suggested comparable in performance to nasopharyngeal samples. [3–7]. However, comparing different sample sites across different studies is difficult owing to the heterogeneity of the studies [8].

## Objectives

Our aim was to ascertain if alternative sampling sites (oropharynx, anterior nasal or saliva) are viable for SARS-CoV-2 detection via RT-PCR in comparison to nasopharyngeal swab, and if self-collection of anterior nasal samples is feasible.

## Study design

### Patients

The study population consisted of 43 outpatients with COVID-19 and upper respiratory symptoms including loss of smell [mean age 40 years (age range 20–70 years); 19 males], who took part in CANNAS-study from January 15th to March 25th, 2021 (R20090, NCT04728919). All patients were RT-PCR positive for SARS-CoV-2 from nasopharyngeal swab, and they had been recruited to the study during contact tracing phone call by the infectious diseases unit. The patients were from the Tampere University Hospital region, and they were recruited only if they could arrive to the study visit without exposing others. The exclusion criteria were the absence of upper respiratory symptoms, onset of symptoms over ten days ago, pregnancy or breastfeeding, and previous vaccination against COVID-19. The duration of symptoms until sample collection was on average 5.4 days (1.82 days SD).

### Sampling and sample preparation

Four sampling sites per patient were selected for the study: nasopharynx, oropharynx, anterior nasal and spat saliva. Nasopharyngeal and oropharyngeal samples were collected by the same physician (D.K.) following established sample collecting guidelines. Nasopharyngeal swab samples were collected unilaterally with FLOQSwabs 503CS01 (Copan Italia SpA, Brescia, Italy) flocked swabs. Oropharyngeal samples were collected with FLOQSwabs 502CS01 (Copan Italia SpA, Brescia, Italy) or HydraFlock 25-3506-H (Puritan Medical Products Company LLC, Guilford, USA) flocked swabs.

The anterior nasal samples were collected by the patients themselves with FLOQSwabs 502CS01 or HydraFlock 25-3506-H flocked swabs. The sampling site in this study refers to the area behind the nasal vestibulum, confined by medial and superior aspect of the inferior turbinate, the nasal septum, anterior aspect of the middle turbinate and the dorsum. The same swab was applied to both nares and the patient was instructed to rotate the swab at the site for 10 seconds.

For saliva samples, patients were asked to clear their throat and then proceed to collect saliva in their mouths for ten seconds. Saliva was spat into a 50-millilitre conical-bottom polypropylene centrifuge tube. The collection was repeated three times or until the bottom cone of the tube was full (approximately four millilitres).

When self-collecting samples (saliva and anterior nasal), patients followed written and illustrated instructions under physician supervision. The physician did not need to intervene in the sample self-collection process as no safety issues were encountered.

The transport medium used for the swab samples was UTM Mini 3 ml 353C (Copan Italia SpA, Brescia, Italy). Swab sample transport medium was analysed undiluted whereas all saliva samples were diluted with phosphate buffered saline with a ratio of 1:1 regardless of the sample volume collected to reduce sample viscosity.

## RT-PCR

The analysis of all the samples was performed at Fimlab Laboratories alongside regular COVID-19-samples with standard operating procedures and personnel with Allplex™ 2019-nCoV Assay (Seegene Inc, Seoul, South Korea) RT-PCR for SARS-CoV-2 with three target genes (N-, Rdrp- and E-genes). The cycle threshold (Ct) values of each target were given by the RT-PCR protocol. A sample site was defined positive if at least one target gene other than E had Ct value $\leq 40$. All tests were done in accordance with the manufacturer's instructions.

## Statistical analysis

Positivity of oropharyngeal, anterior nasal and saliva samples were compared with nasopharyngeal samples by using Fisher's exact test. An arbitrary Ct value of 40.01 was given for each negative target gene result (n = 11), and the mean of the three different target values was used for the analyses. The median Ct values of oropharyngeal, anterior nasal and saliva samples were compared to those of nasopharyngeal samples by Related-Samples Friedman's Two-Way Analysis of Variance by Ranks. The correlations between Ct values and duration of symptoms were calculated using Spearman's rank correlation. The statistical analysis was performed with SPSS Statistics 27.0.1.0 (International Business Machines Corporation (IBM), Armonk, USA).

## Ethical statement

The study was approved by the Tampere University Hospital Ethical committee (R20090, registered at clinicaltrials.gov NCT04728919) and all the patients provided written informed consent.

## Results

Out of the 43 patients with positive nasopharyngeal sample, all gave a positive anterior nasal and saliva samples while two oropharyngeal samples were negative (Table 1, p = 0.49; Fisher's

**Table 1. SARS-CoV-2 RT-PCR results and correlations according to sampling site.**

| | | | Pooled Ct from 3 gene targets | | Rank correlation to nasopharynx | |
|---|---|---|---|---|---|---|
| | **Positive** | | **Median (IQR)** | **P-value, compared to nasopharynx[a]** | **Rho** | **P-value** |
| Sampling site | | | | | | |
| Nasopharynx | 43 | 100% | 22.6 (21.2 to 25.1) | | | |
| Anterior nasal | 43 | 100% | 24.3 (20.8 to 29.8) | 0.145 | 0.717 | <0.001 |
| Oropharynx | 41 | 95.3% | 25.7 (23.2 to 27.9) | <0.001 | 0.670 | <0.001 |
| Saliva | 43 | 100% | 26.7 (22.8 to 29.8) | 0.007 | 0.413 | 0.006 |

Ct, cycle threshold; IQR, Inter-quartile range.

[a]Related-Samples Friedman's Two-Way Analysis of Variance by Ranks with Bonferroni correction. Three gene targets (E-, RdRp- and N-gene) received 504 results from 172 samples. To allow these statistical comparisons, each negative target gene result was given.

exact test). There were no statistically significant differences between the median Ct values of self-collected anterior nasal cavity samples when compared to the nasopharynx. However, the median Ct values of oropharyngeal and saliva samples were higher in comparison to nasopharyngeal samples (Table 1, Fig 1). The two self-collected sample types, anterior nasal cavity and saliva showed no statistically significant difference in Ct values (p = 0.316). The Ct values of anterior nasal, oropharyngeal and saliva samples correlated significantly with those of the nasopharyngeal in the whole group level.

## Discussion

Previous studies have identified saliva [9] and anterior nasal swabs [10] as viable alternatives to nasopharyngeal swab in SARS-CoV-2-diagnosis. The current results are in line with this. The

**Fig 1. Patient sample Ct values across different sampling sites.** ND, not detected; NPS, nasopharyngeal swab; Ct, cycle threshold.

less invasive, self-collected anterior nasal swab was comparable to nasopharyngeal swab in symptomatic and relatively high viral load patients: all positive cases were found with no significant difference in median Ct values. Saliva samples also found all positives, but at higher Ct values. Anterior nasal samples had wider inter-quartile ranges in Ct values, which we assume to result from the self-collection of samples. Confounding factors were controlled by using the same type of swab device in certain anatomical site, same transport medium across all swab samples, having the same person collecting all the nasopharyngeal and oropharyngeal samples and using the same RT-PCR assay and laboratory.

Pandemic has continued for over a year, refusal of testing seems to be emerging as a new challenge [11]. Providing a possibility of a less invasive sample collection method might be helpful in improving test compliance. Alternative testing sites may also be needed e.g. for people with coagulation disorders predisposed to bleeding or for children. Even though nasopharyngeal sampling is safe, serious complications can occur [12] and compared to the nasopharynx, anterior nasal cavity is easier to access if local treatment of sampling site trauma is needed. Also, self-collected samples open additional avenues to improving testing coverage and by implementing controlled self-testing, the need of protective equipment and medical staff is reduced, and patients might be more willing to get themselves tested if the sample was self-collected. The challenge, though, is maintaining high standards of testing so that results are still reliable.

## Study limitations

The study included 43 SARS-CoV-2 RT-PCR positive outpatients. Confounding factors were carefully controlled but due to the sample size we cannot rule out the possibility of minor differences. In addition, our study did not recruit any asymptomatic patients so the behaviour of their samples could differ from those of symptomatic patients.

## Acknowledgments

The authors would like to thank study nurses Merja Rumpunen and Tiina Mäki as well as the contact tracing unit from Tampere University Hospital for patient management and guidance. The authors would also like to thank statistician Mika Helminen from Tampere University Hospital for his advice on statistical analysis.

## Author Contributions

**Conceptualization:** Dominik Kerimov, Pekka Tamminen, Hanna Viskari, Lauri Lehtimäki, Janne Aittoniemi.

**Data curation:** Pekka Tamminen.

**Formal analysis:** Pekka Tamminen, Lauri Lehtimäki, Janne Aittoniemi.

**Funding acquisition:** Lauri Lehtimäki.

**Investigation:** Dominik Kerimov, Pekka Tamminen, Hanna Viskari.

**Methodology:** Pekka Tamminen, Lauri Lehtimäki.

**Project administration:** Lauri Lehtimäki.

**Resources:** Dominik Kerimov.

**Supervision:** Lauri Lehtimäki, Janne Aittoniemi.

**Validation:** Lauri Lehtimäki.

**Writing – original draft:** Dominik Kerimov, Pekka Tamminen.

**Writing – review & editing:** Dominik Kerimov, Pekka Tamminen, Hanna Viskari, Lauri Lehtimäki, Janne Aittoniemi.

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
