## [Decision Letter · Decision Letter 0]

22 Sep 2021

PONE-D-21-22703

Sampling site for SARS-CoV-2 RT-PCR - an intrapatient four-site comparison from Tampere, Finland

PLOS ONE

Dear Dr. Kerimov,

Thank you for submitting your manuscript to PLOS ONE. After careful consideration, we feel that it has merit but does not fully meet PLOS ONE’s publication criteria as it currently stands. Therefore, we invite you to submit a revised version of the manuscript that addresses the points raised during the review process.

Please revise accordingly.

We look forward to receiving your revised manuscript.

Kind regards,

Academic Editor

PLOS ONE

“This work was supported by funding from Tampere Tuberculosis Foundation; Research Foundation of the Pulmonary Diseases; Competitive State Research Financing of the Expert Responsibility area of Tampere University Hospital; Tampere University Hospital Support Foundation. Funders did not have any role in any part of the study.”

Funding information should not appear in the Acknowledgments section or other areas of your manuscript. We will only publish funding information present in the Funding Statement section of the online submission form.

“The funders had no role in study design, data collection and analysis, decision to publish, or preparation of the manuscript”

Reviewers' comments:

Reviewer's Responses to Questions

**Comments to the Author**

1. Is the manuscript technically sound, and do the data support the conclusions?

Reviewer #1: Yes

Reviewer #2: Yes

Reviewer #3: Partly

2. Has the statistical analysis been performed appropriately and rigorously? 

Reviewer #1: Yes

Reviewer #2: Yes

Reviewer #3: I Don't Know

3. Have the authors made all data underlying the findings in their manuscript fully available?

Reviewer #1: Yes

Reviewer #2: Yes

Reviewer #3: Yes

4. Is the manuscript presented in an intelligible fashion and written in standard English?

Reviewer #1: Yes

Reviewer #2: Yes

Reviewer #3: No

5. Review Comments to the Author

Reviewer #1: The manuscript is a short report on a very important study of the importance of the sampling site for diagnosis of SARS- CoV 2. Studies like this should have been done in the beginning of the pandemic and the actual study adds important information. For the standard diagnostics with nasopharyngeal samples trained health care workers are needed and because of discomfort several patients, especially children, don’t accepted repeated examinations.

The actual study is well done and presented. That anterior nasal swabs perform as good as nasopharynx swabs is an important finding.

The material is small with only 43 diagnosed positive patients and small differences could have been missed. I recommend to add a sentence about this limitation. At all places I recommend to instead of “no difference” write “ no statistically significant difference” .

I recommend to remove row 129-131 “No significant correlation was found between patient age or duration of symptoms and Ct values of any sampling site. No statistically significant difference was found in chronic disease, allergies, smoking, body temperature or symptoms between sexes”. It is not likely there was enough statistical power to detect significant differences in these subgroup-analyses in such a small patient group.

Reviewer #2: The manuscript by Kerimov et al describes results from SARSCoV2 PCR in samples from different sampling sites. The main finding is that samples from anterior nasal cavity are as good as nasopharyngeal samples for detection of SARSCoV2. This is in line with previous reports.

The manuscript is concise and well written.

Major comments;

My major concern is the small sample size and the fact that all patients were symptomatic, presumably with a short duration of symptoms before sampling. This must be discussed in the discussion section, ie adding a paragraph on limitations of the study. Futhermore no patients with low or no virsl load was included in the study.

The conclusion must be corrected accordingly, anterior nasal cavity samples yields comparable results to nasopharyngeal samples in symptomatic patients with rather high viral load.

Minor comments;

Line 125 is this correct? Should it read higher ct values instead?

Reviewer #3: I welcome to review this paper on appropriate sampling sites for RT-PCR Sars-Cov-2. A highly interesting topic during the present pandemic. The authors show that sampling in the upper airways could be done at different sampling sites (anterior nose, oropharynx and saliva) with satisfactory result compared to the established sampling site nasopharyx. This could be more convenient and enhance sampling frequency in some patient groups. But, the major drawback of the present study is the number of the studied patient cohort, only 43 patients. Due to this small number the conclusion will undeniable be vague. Other sampling sites than nasopharynx in the upper airways could be used, but the sensitivity is reduced and this reduced sensitivity, the magnitude, could not be assessed in this study, due to the low number of observations. Minor concerns are: the authors don't specify which symptoms qualify for inclusion in the study. For the sampling of oropharynx and anterior nose two different testing kits are in use, with no information on comparing prestanda. The target genes for RT-PCR is only presented in a footnote in a table, a more vigorous description is required, and also the reason for pooling the  resulting means of three different genes for each sampling site, and than comparing the medians of this operation. A description and discussion of the problem with the sampling volume is lacking.

6. PLOS authors have the option to publish the peer review history of their article (what does this mean?). If published, this will include your full peer review and any attached files.

Reviewer #1: **Yes: **Rune Andersson

Reviewer #2: No

Reviewer #3: **Yes: **Gunnar Jacobsson

---

## [Author Response · Author response to Decision Letter 0]

23 Oct 2021

Reviewer #1: 

We have added a paragraph about sample size limitation in the text, starting from line 134. 

We have changed all the phrases “no difference” to “no statistically significant difference”. 

We have removed lines originally at 129 to 131. 

Reviewer #2: 

Regarding the question about a possible error at line originally at 125. Yes, it should read “higher” rather than “lower” and it is now corrected. 

We have added a paragraph about sample size limitation in the text, starting from line 134. In addition, we have added a mention in the discussion about the patients being symptomatic and of relatively high viral load. 

We have changed the conclusion sentence to emphasize the symptomatic patients, line 116. 

Reviewer #3: 

We have added a paragraph about sample size limitation in the text, starting from line 134. 

We have added sentence about the inclusion symptoms for the patients to the manuscript, line 51. The patients had to have had upper respiratory symptoms including loss of smell. 

Two different testing swabs for the oropharynx and anterior nasal samples were used. These swabs were macroscopically identical flocked nylon swabs with a thicker handle compared to the nasopharyngeal swabs, for ergonomic and safety reasons. Both are from very reputable manufacturers, and we use these swabs from both manufacturers for PCR samples at our lab successfully so we felt that it would have been extraneous, within the scope of this study, to first asses their real-world performance on patients. 

We have added text about the target genes, line 87. 

The Ct values of each target gene were pooled to a mean Ct value per sample to enable comparison between patients, as a sample can be positive with one, two or three genes. The Ct value deviations between different genes within the sample were also small.

---

## [Decision Letter · Decision Letter 1]

4 Nov 2021

Sampling site for SARS-CoV-2 RT-PCR - an intrapatient four-site comparison from Tampere, Finland

PONE-D-21-22703R1

Dear Dr. Kerimov,

We’re pleased to inform you that your manuscript has been judged scientifically suitable for publication and will be formally accepted for publication once it meets all outstanding technical requirements.

Kind regards,

Academic Editor

PLOS ONE

Additional Editor Comments (optional):

Reviewers' comments:

Reviewer's Responses to Questions

**Comments to the Author**

1. If the authors have adequately addressed your comments raised in a previous round of review and you feel that this manuscript is now acceptable for publication, you may indicate that here to bypass the “Comments to the Author” section, enter your conflict of interest statement in the “Confidential to Editor” section, and submit your "Accept" recommendation.

Reviewer #1: All comments have been addressed

Reviewer #2: All comments have been addressed

2. Is the manuscript technically sound, and do the data support the conclusions?

Reviewer #1: Yes

Reviewer #2: Yes

3. Has the statistical analysis been performed appropriately and rigorously? 

Reviewer #1: Yes

Reviewer #2: Yes

4. Have the authors made all data underlying the findings in their manuscript fully available?

Reviewer #1: No

Reviewer #2: Yes

5. Is the manuscript presented in an intelligible fashion and written in standard English?

Reviewer #1: Yes

Reviewer #2: Yes

6. Review Comments to the Author

Reviewer #1: (No Response)

Reviewer #2: All comments have been adressed.

Minor :

Under limitations: the authors should consider rephrasing the sentence stating that samples from asymptomatic patients were not included. Behave is probably not the right word to use.

7. PLOS authors have the option to publish the peer review history of their article (what does this mean?). If published, this will include your full peer review and any attached files.

Reviewer #1: **Yes: **Rune Andersson

Reviewer #2: No

---

## [Editor Report · Acceptance letter]

8 Nov 2021

PONE-D-21-22703R1 

Sampling site for SARS-CoV-2 RT-PCR - an intrapatient four-site comparison from Tampere, Finland 

Dear Dr. Kerimov:

I'm pleased to inform you that your manuscript has been deemed suitable for publication in PLOS ONE. Congratulations! Your manuscript is now with our production department. 

Kind regards, 

on behalf of

Dr. Robert Jeenchen Chen 

Academic Editor

PLOS ONE